# Engagement and Retention of Families in Universal Australian Nurse-Home-Visiting Services: A Mixed-Methods Study

**DOI:** 10.3390/ijerph20156472

**Published:** 2023-07-28

**Authors:** Belinda Mawhinney, Jennifer A. Fraser

**Affiliations:** Sydney Nursing School, University of Sydney, Susan Wakhil Health Building, Camperdown, NSW 2050, Australia; jennifer.fraser@sydney.edu.au

**Keywords:** nursing practice, engagement, retention, multilevel mixed methods, child and maternal health, child and family health services, nurse home visiting, universal health services, nurse–family relationship, families with complex needs, child protection

## Abstract

Family support is offered to Australian parents of young children using a mix of targeted and universal child and family health services including nurse-home-visiting programmes. These rely on the voluntary engagement of families. In this study, the capacity to engage and retain families, including those at risk of becoming involved with child protection services, was examined. The broad objective was to identify nursing practices used at the interface of health and child protection services and to articulate those practices. Child and Family Health Nurses (CFHN) (*n* = 129) participated in a pragmatic, multilevel mixed-methods study. A questionnaire was used to identify nursing practices in the first phase of this study followed by focus groups in the second phase to describe these practices in more detail. Three practice themes were identified and described: enrolment, retention and conclusion of the nurse–family relationship. Universal child and family health services feature flexible, advanced, and multidimensional family support services including child protection practices. This paper focuses on practices employed by nurses to engage and retain families where child protection concerns are identified.

## 1. Introduction

Nurse-home-visiting programmes for families with young children are key to the public health approach to child protection in Australia. Child and Family Health Nurses (CFHN) are organised into models of both targeted nurse home visitation services for early intervention of child abuse and neglect and universal services for prevention. The work involves providing ongoing support for families with young children (aged 0 to 5 years), making risk assessments, referrals, and where there is a significant risk of harm to the child, a report to child protection authorities as required by state and territory laws [1]. The model of care is best described as progressive universalism [2]. This is not a straight application of universal health care. Progressive universalism means nurses adjust the dosage of contact based on the level and complexity of need and vulnerability identified in a family. The expectations of contemporary nursing practices mean nurses need to work to prevent child maltreatment [3,4,5,6] as well as work in partnership with families to meet their parenting goals and aspirations under duress.

However, as the systemic gaps in the child protection system continue to widen, frontline clinicians in these services, predominantly nurses, are required to fill these gaps. This is a complex practice landscape for these health professionals. They can be ill prepared for, and do not recognise themselves as, child protection experts. Whilst the goal of their work with families is aligned to statutory child protection casework, that being to protect children from harm, there are vast differences across professional standards for this professional group, including the way in which risk is assessed and how it is differentiated from safety [7].

The effort to maintain relationships with families and to retain them in these services is very challenging. Where suspected child maltreatment is reported to the statutory child protection agency, a response does not automatically follow. Consequently, progressive universalism enables nursing practice to be extended to the point of a tertiary response without the sanction. Although nurses are encountering the practice challenges associated with this functional role change, little evidence exists to explain how this has changed nursing practice.

The goals of the universal nurse-home-visiting programmes are to provide access to universal screening and are intended to offer support that is based on health and psychosocial assessments. Studies are progressively placing greater emphasis to describe contemporary practices and standards across nursing services [8,9,10]. A qualitative study [11] with Australian nurses (*n* = 21) from various clinical contexts, though all worked directly with children. The study found challenges in the accurate identification of maltreatment, the influence of personal values and beliefs of nurse, as well as the impact of diverse cultural practices with children. These are valuable insights into nursing practice, though considerable gaps in knowledge continue to exist that extends these ideas and identify the specific nursing practices in this context.

In recent years, some international research attention has been given to responding to child protection needs in the context of universal nursing services [12]. Yet, uncertainty prevails about how best to effectively support families involved with child protection services or who are at risk of becoming involved. The relationship between nurses and mothers has benefited from research focused on establishing and maintaining the relationship between nurses and mothers in the context of nurse-home-visiting models [13]. However, adding the legal and policy mandate of child protection adds a further layer of complexity that requires a greater understanding [7]. Some studies have focused on the client perspective to identify the personal attributes required by nurses to establish a positive working relationship. In a Canadian study [14] clients reported their preference for non-judgmental, friendly and honest professionals. Trust was also identified as a core element of the working relationship, particularly where the relationship was tested by a client’s history of broken trust with service providers. The working relationship between nurse and family needs to be robust enough to tolerate the conversations necessary to complete screening around maltreatment risks. Practical strategies, such as the SPIKES protocol encouraged health professionals to consider the elements of Setting, Perception, Invitation, Knowledge, Emotional, Strategy and Summary when framing a conversation about mandatory reporting [15]. Though in reality, when faced with child protection concerns, nurses must employ advanced practice skills beyond the initial explanation about being a mandatory reporter. Their approach to practice, the nurse’s personal attributes and the quality of engagement between the nurse and family are just some examples of influences on nursing intervention and the outcomes that follow. CFHNs must reach these families to both engage and retain them to benefit from existing services. Further development of evidence-based practice standards requires urgent research attention to navigate this complex area of practice.

Studies have been designed to unravel the complexities of the nurse—family relationship outside of dedicated services tasked with the prevention of child abuse, such as the Nurse Family Partnership model. A study conducted in Sweden [12] conducted a study in Sweden which highlighted the importance of rapport when addressing sensitive topics, such as the risk of maltreatment. However, their study found that practice was underpinned by experience, rather than being addressed in formal education or training. Empirical evidence has established common influences on maternal engagement [16] and the nursing profession has made significant advancements in the past two decades to redefine the scope of their practice beyond the biomedical model to target social and cultural determinants of health and the complexity of family violence. It is timely to now further support this practice evolution by delving deeper into the practices used when working with families with complex needs.

What remains an ongoing challenge for nurses working in voluntary service models is the element of choice to engage is just that, a choice. Families are free to choose whether or not to accept this service [17]. While CFHN are responsible for delivering a public health approach to children 0 to 5 years, the implications to practice are subtle. Simply put, the service is accessible to families but what specific practice is required when the family chooses to cease engagement and child maltreatment concerns are present. If the family chose to disengage, is it enough to respect their self-determination? Or is further action needed to flag the child maltreatment concerns alongside the absence of universal health service? This scenario provides some preliminary considerations of the complex landscape nurses work within when working with families where forms of family violence including child abuse are used.

Considering the evolution of these practice changes experienced in recent years, and in consideration of the context described above, a pragmatic, mixed-methods study was conducted. The purpose of our study was to explore the practice implications of this complex landscape among a cohort of CFHN in an Australian metropolitan setting. The relationship between a nurse and a family was the primary focus, particularly during the retention phase.

## 2. Background to the Study Design

The approach used for this study was based on two key considerations. The first related to the change in direction of service delivery to families preceding this study. Legislative reform had resulted in a shift in roles and responsibilities for services to augment the work undertaken with families to divert the need for statutory intervention. The second related to fundamental principles underpinning child and family health nursing. That is, the formation of a nurse–family partnership and how this relationship unfolds across time. This study first looked at how nurses were reporting their knowledge, confidence and practices following on from changes to policy prompted by the reform. Next, the impact on appropriately managing child maltreatment risk in the context of being a service operating on voluntary terms was explored. The aim was to examine how voluntary engagement aligns with the function of being in a supporting role for families when disengagement occurs and risks exist. Based on the literature review above as well as these background contextual considerations, the following research questions were posed for this study:


**Phase One Research Questions**


How knowledgeable are CFHNs in relation to their mandatory CAN reporting responsibilities?What are the practice responses of CFHNs when managing families with complex service needs?How do CFHNs engage and respond to families with complex service needs?What is the role of education and training in preparing CFHNs for complex care offered to vulnerable families?


**Phase Two Research Questions**


Do CFHN consult with colleagues when reporting child protection concerns?How are child protection concerns raised with the family?Does disclosure of concern for children affect continued engagement with the family?How do CFHN support families once child protection concerns are raised?How are families discharged from the service once they cease to engage?

## 3. Methods

**Participants:** CFHN (*n* = 129) were recruited to participate in this study. The sample of participants worked in two neighbouring workforces in metropolitan Sydney. Therefore, eligibility included all CFHN, including nurses working in the universal or sustained health home visiting programs, along with the Nurse Unit Managers, Clinical Nurse Educators and Clinical Nurse Consultants. Participants were required to be knowledgeable and insightful about the topic, in addition to being available and motivated to articulate a self-reflective account of practice [18]. The sample was mostly women aged over 50 years with more than 20 years practice experience. The sample was considered representative CFHN in NSW based on workforce data [19] and other studies at the time [20].

Ethics approval was granted by the Royal Prince Alfred Hospital Ethics Committee in October 2014. Additional approval was granted by South Western Sydney Local Health District Ethics Committee to include all three community health service centres used for data collection in phase one and two.

### 3.1. Phase One: Approach to Quantitative Data Collection

**Procedures:** With the approval and support of senior managers, all staff within each workforce group were invited to participate in this study. Eligibility for phase one was based on current employment in the two participating services. The questionnaire was tested using a pilot group of five frontline nurses from the participating workforces. The pilot test provided feedback to content leading to improvements in clarity of questions and provided a benchmark for time required to complete the questionnaire (45 min).

Phase one data collection occurred within designated professional development meetings between February and March 2015. Participant information sheets and consent forms to participate in phase two were also issued. Any staff absent from the meeting were supplied with a copy of the research information pack, including a copy of the nine-page questionnaire, participant information sheets for both phases, a consent form to participate in phase two and an addressed envelope to return to the primary author. An additional 41 information packs were issued to managers and educators for distribution to nurses absent from the designated meetings used for data collection.

### 3.2. Phase Two: Approach to Qualitative Data Collection

**Procedures:** Purposive sampling was used in phase two with all participants having also participated in phase one. All participants in phase one were eligible to consent to participate in phase two. CFHNs consenting to participate in phase two were subsequently invited via email to attend the focus groups. Of the 39 consenting participants, 27 participated in total based on availability to attend the focus groups. Three focus groups were conducted. Two focus groups were held for frontline nurses (one per workforce) and a combined group for all Nurse Unit Managers (from both workforces).

### 3.3. Approach to Data Analysis

Multilevel mixed-methods is a contemporary approach to the mixed-methods methodology [21]. This approach was used to address practice on multiple levels as well as explore the relationship between these levels. Nursing practice was explored on two levels, individual and team-based practice. Individual practice was assessed using the 81-item questionnaire in phase one. Team based practice, such as case reviews and group supervision, were the subject of the focus groups in phase two.

The stage at which mixing occurs in mixed methods is the subject of debate [22]. Based on a study of mixed-methods research [22] the argument is made that mixing can occur at multiple stages. Mixed methods were used during phase one, with the questionnaire using both binary (Likert Scale responses) and open-ended text items. Mixing was then used in phase two as focus group participants discussed quantitative results and interpreted emerging practice concepts. As participants examined both integration and variation of practices, the repeated application of mixing data occurred as this study aimed to explore both homogeneous and heterogenous practices. Mixing then occurred during the analysis and integration of both quantitative and qualitative data.

An interpretive analysis [23] was applied to the data from both phases using the Integrated Theory of Parent Involvement [24]. This conceptual model argues parents move through three distinct phases when accessing home visiting services. First, intent to enroll with a service, then enrolment followed by retention. Within these three phases, four consistent influences are considered. That being individual factors, provider factors, programme factors and systemic drivers. Provider factors were used as a lens to report results from this study.

## 4. Results

The focus of this paper is engagement, moreover the retention of families where nurses have identified risk of family violence. Results are presented based on applying the Framework of Practice for working with families with multiple and complex needs [25]—(see Figure 1). The framework depicts the additional factors related to individuals, programs and systems which framed the broader study from which this paper is based. This paper specifically takes the perspective of the provider with results presented across the enrolment, retention and conclusion phases. The framework argues the critical phases of the relationship between nurse and family exist initially at enrolment—where families are engaged, followed by retention—where families continue to access care and finally, in the conclusion phase—which can occur either with or without warning. This framework is an extension of the Integrated Theory of Parent Involvement [24] arguing for a final stage of parental involvement that occurs during the conclusion phase. The vertical line in Figure 1 shows how the extended model aligns with the McCurdy and Daro model. Results are presented according to phase one and phase two of this study.

### 4.1. Phase One Results

All participants knew of the statutory requirement to report concerns, with most (93%) having had reporting experience. A smaller proportion (15%) also reported notifying concerns with their Nurse Unit Manager (NUM). A similar proportion (17%) admitted having suspected but not reported child maltreatment. Using open text responses, analysis of the reasons for not reporting maltreatment concerns was (a) lack of experience or confidence (5%); (b) compliance with government policy (i.e., the decision-making tool recommended a report was not required) (4%) and (c) fear of consequence from the family (2%). No participants reported the practice of discussing concerns directly with the family as a reason for not reporting maltreatment.

A confidence scale was developed to assess confidence across a range of practices, including the identification of abuse types (physical, emotional, neglect, sexual, and domestic violence); to determine whether a concern met the risk of significant harm threshold; to report child maltreatment; and to manage ongoing care to the family once maltreatment has been determined. Each item was measured using a Likert Scale of 1 (Extremely Not Confident) to 7 (Extremely Confident). The mean confidence scale was found to be 5 or more (this equated to ‘I am confident’ on the Likert Scale used in the questionnaire). Completion of the Domestic Violence Routine Screening tool was found to be the most confident practice amongst participants (M = 5.76, SD = 1.19).

When it came to working with families with complex needs, more than half (67%) reported confidence in this area of practice. Almost three-quarters (72%) provided an open text response to describe the management of suspected child maltreatment. Eighteen different practice responses were described in the act of managing care, with the most frequently reported practices including reporting concerns, referrals to support services and consultation with a manager or peer. Engagement was reported as a practice response by almost one-third (27%), rating this as the fourth most frequent practice. One the least frequent practices was working in partnership with the family, which was reported by only one participant. The practice CFHNs had the least confidence (M = 4.87, SD = 1.15) in was once a child was suspected to have been abused or neglected.

A scale was also developed to examine the frequency of nursing practices. Similar to the confidence items, the frequency items ranged from 1 (never) to 7 (always). Practices included applying professional judgement to decision making; consultation with other professionals who work with children; consultation with NUM; consultation with peer; application of the online decision-making tool (Mandatory Reporter Guide); follow health policy; contact Health Child Wellbeing Unit (CWU); report to statutory child protection agency; implement knowledge from professional development and training; discuss family in clinical group supervision; present family at case review meeting; refer to Family Referral Service; make an additional child protection report (where required). Most behaviours were found to have a frequency score of 5 (very frequently on the questionnaire Likert Scale). Behaviour included policy compliance as highest frequency (M = 5.45, SD = 1.59), and the lowest reported behaviour was referring to a Family Referral Service for family support (M = 3.75, SD = 1.54). Overall, nurses reported to deploy the listed practices very frequently (Total frequency M = 4.92, SD = 1.11).

A range of homogenous practices were found. Using two separate items, participants reported the frequency using the same 7-point Likert Scale to measure the frequency of practices used when families are engaged, followed by an item that asked participants to report the frequency of practice when families were not engaged in the service. Seven practices were included in both items, those being refer concerns to the CWU; report concerns to the statutory child protection agency; apply Family Partnerships principles to practice; work collaboratively with other worker or agency; discuss family at case review meeting; discuss family at clinical group supervision; discuss family with peer or NUM and increase attempts to contact the family. Common practices included working in a family partnership informed approach (engaged 69% versus non-engaged 61%) and increased contact with the family (engaged 59% versus non-engaged 52%). Participants were more likely to report consultation with a manager or peer when the family was engaged (engaged 71% versus non-engaged 68%) and more likely to present an engaged family at case review (engaged 61% versus non-engaged 56%). Meaning, families at risk of disengagement from the service were less likely to receive consultation or review compared to those families actively engaged in the CFHNS. With a focus on the variations in practice responses for families not engaged with the service, two specific practices were included which related to application of the organisational policy “failure to attend” and engaging with another discipline to complete a joint assessment. Half the participants reported applying this policy as the most frequent practice in these circumstances. Whereas participants were found to be much less inclined to discuss a family at clinical supervision. Overall, no significant difference was found between practices used with engaged versus non-engaged families.

An exploration of increased contact with a family occurred through open text responses. The majority (94%) of participants provided a response to this item. Intervention, such as assessment, engagement and support, was reported by three-quarters (77%) of participants. Increased contact for the purpose of surveillance was reported by almost half (40%) of the sample, with examples of practice intentions including monitoring and observing. When describing ideal practice with families, descriptors of individual practice was provided by a small number (15%) of participants. Examples included being confident, approachable, and knowledgeable. Participants also provided open text responses about practice barriers encountered when working with disengaged families. Responses were categorised into three themes: collaboration (34%), systems improvements (25%) and individual practices (22%). Examples of practice barriers in collaboration included the family’s willingness to engage. Individual practice barriers included practitioner confidence, divergent practices used for contact and finding the careful balance between multiple contact attempts against the choice of the family to accept or decline services.

An additional item was included to seek participant views about barriers to decision making. Participants were asked “what factors or circumstances make it difficult for you to make decisions when working with vulnerable families?”. Responses were categorised into the same three themes used with practice barriers. Barriers to decision making included collaboration (58%), individual practices (46%) and system improvements (25%). Participants cited the willingness of a family to engage as an indication of a collaborative barrier and tenuous engagement with family as an individual practice barrier.

### 4.2. Phase Two Results

Family violence assessment during initial engagement with families was problematic. Without an established nurse–family relationship, families are not willing to disclose violence. At the same time, there is benefit in asking questions about family violence. Identification and response to risk factors is seen as the CFHN’s role by many families in Australia. “*It is much more comfortable to be able to tell someone something like that if you’ve already mentioned prior your, at, say the first visit about confidentiality and your duty of care that whatever they say is in confidence except if there’s a serious safety concern then it’s you know, you are a mandatory reporter*” (participant, focus group 1). Another nurse explained “*they tell you when they’re ready*” (participant, focus group 3). Being able to effectively engage a family in the service or programme, will predict retention and the participants spoke of being clear about their professional requirement to safeguard children during the enrolment phase. That it is one way to reinforce this message. Being clear early on allows families to make informed decisions about disclosing the presence of risk. One participant clarified that without an established working relationship, families may be less likely to disclose violence.

In the second phase of this study, we explored the developing nurse–family relationship. Speaking openly and honestly during enrolment was thought to be critical in the formation of this relationship.

“*Honesty is vital, like if they are honest with the families they tell them, ‘I’m the mandatory reporter, this is what this means’, whatever, and yeah, with vulnerable families they have an honest conversation and very rarely will those families disengage because I think they value that honesty*”(participant, focus group 2).

Experienced nurses talked about changing practices in relation to discussing risks and their duty to report concerns. Standard practice now includes reporting these concerns to the senior nursing staff member, the Nurse Unit Manger (NUM). The purpose of consulting with a NUM was to guarantee that the manager had oversight of the complexity of the work and that they were able to support and direct the nurse responsible for working with the family.

“*It just became part of our practice in our team to run it past the NUM, have a discussion with her about making any notification or any phone calls to CWU as to why, what their plan is, what the nurse’s plan is to follow-up and what maybe is the outcome from the phone conversation that they had as well*”(participant, focus group 3).

Focus groups explored how nurses are able to retain families in their programme, even when risks were identified and discussed with the family. Personal attributes such as being confident were viewed as being essential to allow families to remain engaged. “*It’s just being confident, isn’t it, confident in the way you approach them*” (participant, focus group 2). Being genuine and kind was also seen to be a valuable personal attribute.

The connection between confidence and experience was considered by focus groups through lively debate. Do less experienced nurses require more or less oversight than their more experienced counterparts? Views varied with no consistently held belief. The extent to which nurses were required to seek consultation from their manager also proved to be a divergent point of view. Based on focus group data, it was difficult to determine whether the consultations were motivated by compliance, competence or accountability. Unsurprisingly, the experience of consultations varied across the three focus groups—though the order of time where a consultation was conducted changed, the practice context was consistently anchored to families with cumulative risk factors.

“*I feel it’s more important to make the notification rather than discuss it with my NUM first if my—I’ve done the MRG, I’ve spoke to the Wellbeing Unit and it’s telling me the report should be made, then I’ll make a report. On saying that, I will always discuss it with my manager at some point”*(participant, focus group 1).

Consultation with a manager was explored as an important aspect of professional practice. The reasons behind engaging in consultation ranged from personal preference through to a source of support and beyond, such as for policy compliance. There were varied views about whether the level of risk predicted a consultation, meaning if the risk was considered to be significant then a consultation would occur. This was not considered to be a common practice, as systemic changes meant less significant risks prompted a practice response from nurses.

Nurses who described examples of positive outcomes with families appeared to have greater confidence when compared with participants with less experience. Discussions highlighted that confidence can be promoted when nurses receive reassurance, support or coaching. Nurses who were given an opportunity to observe a nurse with more advanced skills were also valuable strategies to develop nursing practice.

Nurses also reported being open and honest in communicating was appreciated by families. When sharing practice examples, advising families about mandatory reporting responsibilities was included in having these honest conversations.

*“… And I told her I was going to make a report to the Child Wellbeing Unit. Mm, and she seemed to take it quite well”* (participant, focus group 1). Where nurses had retained a family in service delivery, particularly after a child protection report was made, was considered a success nurse intervention. Participants perceived engagement was threatened when a report was made.

The boundaries between nurse and family were also considered as having both a positive and negative influence on the retention of families. Having a need to “fix things” for families was identified as a barrier to retention. Where nurses held an enhanced sense of responsibility, yet also needed to manage a significant caseload of families, the capacity to hold many families over extended time was intended to be supportive. However, this had its limitations. “*Some people’s problems aren’t fixable, sometimes they just are, but sometimes they just need to share what’s happened and just so, I don’t know, share the load type thing that you can’t necessarily fix*…” (participant, focus group 3). Knowing when to cease intervention with families is a complex practice reality for nurses, with some participants offering that knowing when to enact this conclusion is difficult. For some participants, services ceased when families became hard to reach or disengaged, rather than when service objectives had been completed. The capacity to provide ongoing support to families was considered critical in family retention. Beyond the skill of risk identification, participants explained the importance of providing support rather than abruptly ending service delivery.

“*… being able to offer some help, whether it’s a referral to another service or you know ‘I’m going to follow you up’ and having that real genuine ability, genuine-ness and desire to have a relationship with that client…”* (participant, focus group 2). Skills required to support families included establishing a relationship of trust with the family allowed nurses to facilitate disclosure of risk, engage in meaningful conversations and to advocate for families to access support services.

When families disengage prematurely, nurses can experience uncertainty. Although the participants practiced in a service based on voluntary engagement, nurses are required to make some attempt to reach out to families. This is particularly so after risks have been identified. “*So, those really vulnerable families, we probably make more than two phone calls and one letter; we need to make quite a few phone calls; some people would even call around and put a note under the door to see if they were still living there. If there was a FACS worker, we would ring them and see if they had seen them. Like we do make quite a bit of effort if they are a really vulnerable family like, certainly the Nurses will say to me, ‘I can’t find them there’, if not there, I’ll ring them, the Community Services and find out from their worker what’s happened*…” (participant, focus group 2). The feelings of uncertainty for participants were compounded by an expression of genuine concern. Disappointment, sadness and upset were feelings also expressed by participants. “*I think it’s quite difficult for some Nurses, some people are happy like, ‘Phew, I don’t have to bother anymore’, but some are really, ‘I’m so concerned about that family or that kid’, like really, quite sad, yeah, just would really like to get in there and help*” (participant, focus group 2). “*You do feel like a bit of a failure at times*” (participant, focus group 1).

“*This one just— she just vanished…it was one text message to say, ‘No, I’m not available today, I’ve moved house, I don’t know my new address yet’. (Then) Phone number was disconnected, no address, nowhere to go and those ones when there was a lot of things in place… we felt like we were getting somewhere, we felt like there was a little shift…{Researcher: ‘so how did that feel?’} and that felt really awful. That felt really awful*”(participant, focus group 1).

Where nurses had successfully retained a family in a working relationship and families became hard to reach, nurses expressed feelings of anxiety, disappointment, frustration, dread, concern and sense of loss. One nurse explained the emotional response to a premature ending in the nurse–family partnership can be complicated by the unknown, “*It’s the uncertain worry is the thing that sort of gets you*…” (participant, focus group 1). Where nurses are no longer working with a family can also signal the absence of any services monitoring a family with risk factors. “*…It’s that emotion that I think the nurses struggle with is the letting go of those families where they know there is nobody else around. There is no one else keeping an eye on them or that child*” (participant, focus group 2).

Consultation occurred with managers when families disengaged from the service without explanation to enable the nurse to discharge the family from the service. The level of risk did not correlate with the extent of concern expressed by nurses. Rather, focus group participants suggested that parent mental health problems and social isolation were often the cause of greater concern, rather than risk that constituted significant harm. The reason for NUM consultation in these cases was seen to be seeking permission to stop and conclude the service, despite the concern and uncertainty. “*It’s voluntary, time to stop, you’ve done the report, you’ve written everything you need to on (the computer system), you’ve done all the numerous, more contacting than what is actually our policy to do because you’re concerned, and you want to stay engaged with this family. You have done all that is possible in your role, you have done it”* (participant, focus group 3).

## 5. Discussion

This study sought to examine the range of practices used by nurses providing universal nurse-home-visiting programs that have a primary health care function that combines child health and development screening alongside a form of child maltreatment early intervention. This includes for families where forms of family violence including child abuse may already exist. The best evidence for this approach is found in the literature dedicated to sustained programs delivered by experienced, specialist nurses over a two-year timeframe. Limited access and eligibility criteria to specialised sustained health home visiting programs (SHHV) has resulted in families with complex needs accessing universal programs. The CFHN nurses working in universal services have faced increasing demands on their skills to provide an effective response. The literature on the prevention and response to child maltreatment is largely contextualised in SHHV. The challenges associated with servicing families with complex needs from a universal service base warranted attention to enhance responses to families across multiple levels, including systems, services, nursing practice and for the quality of care received by the family at the centre of the response. The current study and the focus of this paper was to examine nursing practice in the context of working with families with complex health and welfare needs.

This study was designed to examine the nursing practice deployed by CFHNs who are also providing an important service to safeguarding children, without the structured programme of sustained health home visiting. These universal nurse-home-visiting programs reach families with complex health and welfare needs as well as responses to child abuse and other forms of family violence. This study found participants were operating from principles of progressive universalism, meaning they are actually providing targeted interventions to meet the complex needs of families where children under 5 years of age are at risk of abuse and neglect. Progressive universalism is grounded in voluntary engagement principles. This fact drives much of the complexity for nurses working with families with complex needs in this model. Our findings highlighted the efforts needed to engage hard to reach families notwithstanding the choice inherent in a voluntary service.

### 5.1. Enrolment Phase: Honesty in Relationships

The CFHN participants in this study had an average of 25 years nursing experience, and 13 years practicing as CFHN. However, experience does not guarantee competency [26] or confidence [27]. Two-thirds of the participants had made a child protection report in the previous 12 months. Experience and knowledge increased confidence in the safeguarding role indicating more certainty and less reluctance than reported elsewhere [28]. When it came to standardisation, there was consistency across their practice. This finding is in contrast to other studies that have found more heterogeneity in reporting practices [29].

Participants reported it was standard practice to discuss family violence and maltreatment concerns with the family. Such conversations are known to be a challenging aspect of practice [30,31,32]. This was further confirmed in this study. The need to be honest, and clear with families about concerns is a well-known barrier to reporting [33,34]. Studies have highlighted concerns at this point of the relationship. Concerns need to be discussed, but at the same time, parental engagement is threatened [7,35,36]. Early conversations with families not only set out the role of the visiting nurse, but also contextualise the safeguarding elements of the role, including the mandate to reportdescribed it as “laying the groundwork”.

In an Australian study of primary health care providers [37] the health professionals regarded making a child protection report as an act of betrayal, rather than a sign of trust. However, recent research interest in this field, including this study herein, indicates that relationships are preserved, and that trust can be extended when honesty is prioritised during the enrolment phase [13]. Importantly, honesty was found in the present study to be vital in this phase of family involvement. Furthermore, families were found to rarely disengage where the nurse was up front about the scope of their safeguarding role.

### 5.2. Retention Phase: The ‘Art of Managing Complexity’

Critical analysis of research on reporting behaviours [10,13,37,38] has provided further insight into this complex area of nursing practice. We sought to address issues identified in the literature through a detailed description of practices used when managing the complex needs of a family, beyond that of reporting risk. The foundation of progressive universalism allows for extending contact with families and is based on the premise that it will meet the families’ specific needs. The present study found nurses commonly increased contact with a family when risk of maltreatment was assessed, reinforcing the principles of progressive universalism exist in this service. Assessment, engagement, and support were commonly seen to the purpose of increased contact. Almost half of participants considered surveillance as a function on increased contact. Although a range of practices were described, participants did not overtly articulate the purpose of intervention was to mitigate or manage the assessed risks nor was confidence a standard trait for the cohort. In fact, one-third of participants did not report feeling confident when managing care once maltreatment was identified. Similarly, engagement was not rated highly as a priority when working with families for one-third of the participants. Despite this, the data are rich in descriptors about the multiple levels of practice that contribute to a maltreatment response. We argue that CFHN are no longer well placed to identify and prevent child maltreatment, they too play a critical role in ‘holding the risk’ with the family as they craft a family focused service response to mitigate risk. Operating from progressive universalism enables nurses to adjust the frequency of contact with families in their care. However, simply increasing contact to families is not enough. Nurses must also craft a purposeful response to child maltreatment from a universal health care service. Nurses in this study were committed to deliver a meaningful service to families and valued access, support and ethical practice. However, where contact was increased the purpose was less clear. There are known tensions between the purpose of nursing intervention being supportive or for the purpose of surveillance [39]. When asked to describe the purpose of increased contact, participants were able to identify support, education, referral to other support services amongst examples. Another example was service retention, which involved the nurse continuing to home visit to monitor the risk. Data analysis was not able to decipher whether nurses spoke honestly and directly with families receiving monitoring, which highlights the possibility that in the absence of this clarity—families may prematurely withdraw from service delivery without a shared understanding about the purpose of frequent home visits. Other studies have also emphasised that families must be clear about the purpose of an intervention [40]. When considering how to effectively retain a family in universal health care beyond the enrolment phase, home-visiting nurses must be clear with families about the purpose of their visits. The risk of disengagement increases when the purpose of intervention is not clearly explained. Prevailing views that see families being described as non-complaint or “failed to attend” miss a critical opportunity to reflect on practice and consider how nursing practice may influence this outcome.

Nurses also expressed concern for their role in surveillance of family relationships, violence and child abuse. They were not at all comfortable with the task of monitoring, confirming this as an aspect of practice most likely to uncover risk for child abuse. The role of these nurses included maintaining contact with families, increasing contact, to ensure families were engaged with a service. Where no services were involved, the nurses held genuine concern about the safety of children and their families. In an article [39] supporting their concern, Irish health visitors were reported to also lack confidence in the wisdom of monitoring and believed that monitoring was not an effective safety and protection strategy.

We found that nurses adjusted the frequency of contact or service “dosage” according to their perception of risk to the welfare of children in the family. One study [28] described nursing practice as ‘dancing around families’ (pg. 2244), whereas this study has argued the dance is far more interactive and intended to retain families by meeting their needs. For example, referring to additional support services. This practice demonstrates nurses must be thorough and adaptable when working with families. Whilst practices considered in isolation do not warrant an advanced skill set, it is the integration of skills that come together to formulate an advanced response. In the enrolment phase, nurses show that they are competent, skilled, and genuinely care about the family. Beyond that, the retention phase of intervention requires nurses to weave appropriate and relevant practices to ensure the management of care is curated to meet the unique needs of each family. Although participants were not as confident when managing care to families known to child protection authorities, this study has made a valuable contribution to the literature by richly describing the range of tasks needed when working with families.

Others have suggested that enrolment of families into a service relies on emotional intelligence and empathy [41]. In the current study, nurses described the need to meet the needs of the family to be able to not only enroll, but to also retain the family in ongoing service delivery. The ongoing emotional implications associated when working with families with complex needs suggests these skills are not limited to the enrolment phase. In fact, we would argue the emotional component of this practice does not conclude until the family ceases engagement—and in some cases, the premature withdrawal from the service does not signify an end to the genuine investment shown by nurses towards some families. In a qualitative study [34] findings stressed that beyond the initial threat to retention attributed to making a child protection report there is an emotional toll experienced by health practitioners. The study [34] described health practitioners as “riding the reaction wave” whilst trying to retain family engagement as the health service endeavours to continue providing health care. Acknowledgment of the emotional element of this practice context was further highlighted in a further finding from the study described as “emotional battleground”. Both the current study and the study [34] recognised the multiple levels that interrelate and influence the practice of frontline health workers who are in the business of keeping people safe, though equally impacted by the execution of this purpose.

The range of practices associated with safeguarding highlighted in this study include risk identification, reporting risk, signposting families for additional support services, and consultation with a manager or peer. This has added to the existing literature which found multidisciplinary and interagency collaboration [42] effective communication skills [31] as essential in the prevention and response to child maltreatment [13,31,32,43,44,45,46]. Collectively, these findings depict the range of skills and practices required to manage ongoing care for vulnerable families. Adding to this is the relationship developed and held between nurse and family. Trust was found to be critical to the relationship in our study and others [47,48].

Practices required to work with families with complex needs requires nurses to both work directly with families, such as risk assessment, discussing concerns and increasing frequency of contact to monitor those concerns, coupled with indirect interventions such as reporting, referring and consultation. The combination of the direct and indirect practices demonstrates the multiple levels of activities required to retain families in ongoing care and service delivery. Although diverse and with varied levels of confidence, most participants in the current study articulated practices required to safeguard children against maltreatment.

### 5.3. Conclusion Phase: Uncertain Endings

The emotional toll associated with working with families was found to have implications in all phases of parent involvement, including when engagement was fractured. Similar to another study [37] the current study found when the working relationship between nurse and family ends without warning the emotional impact on the nurse is evident. The critical importance placed on honesty in the early formation of the relationship has been emphasised and must remain equally important across all phases of intervention. Where services have adopted reporting practices where families disengage from service delivery, nurses must have direct and open conversations in the enrolment phase to ensure families comprehend this as a potential outcome of the cessation of the nursing intervention. Whilst some participants find this report serves as a practice that allows some sense of closure, other participants were burdened with concerns about the families no longer receiving service intervention.

Whilst operating from a service grounded in voluntary engagement, nurses often feel compelled to report to statutory child protection services once the family ends the relationship. The juxtaposition of providing a voluntary service yet reporting when a family chooses to disengage requires further exploration to uncover the purpose of such a practice response. This study found that a statutory response is rarely deployed in these instances. This raises further uncertainty about continuing this practice. Some Australian states can share concerns about families within organisational structures, such as the CWU operating within government health services. It is possible concerns of disengagement may equate to risk of harm, rather than risk of significant harm, which is within the scope of practice for the CWU. Unfortunately, CWU are not resourced for providing a direct service response to families. At the same time, they do play an important role in sharing information between health services to enable comprehensive practice responses from frontline health workers.

We compared nurses’ practices of working with families who were engaged with families not engaged. No statistically significant practice variations were found. Across the range of practices, participants identified consultation, policy compliance, applying professional judgement and reviewing a family amongst peers as a case review meeting occurred with the highest frequency. Conversely, missed opportunities were associated with practices that had only recently been introduced with a change in child protection legislation. Less frequently, practices included using the Family Referral Service, applying an online decision-making tool or referring to the child wellbeing unit.

Not only are CFHNs well positioned to engage and retain families in this universal service, but managers were also noted to play a key role in influencing service delivery. Consultation was found to be part of the support for practice in the current study, a finding echoed in other literature [49]. Whilst consultation is an obligation outlined in organisational policy, nurses in the sample were found to value this interaction and considered it reassuring to their practice. The role of managers is not only pivotal in supporting their teams [50], but also knowing the experiences of nursing intervention allows a greater insight into the families accessing the service [51]. The present study found participants to be engaged in frequent consultation, though little insight was gained about the content or quality of the consultation, nor the outcome for families. Nurses were reluctant to access clinical supervision to support their practice with families. This was an unexpected finding that would benefit from further exploration. Whilst the functions of clinical supervision can vary [52], a greater understanding about the efficacy of supervision [53] is needed—particularly in circumstances where CFHN work with families with complex needs from a universal health service. Most, if not all, CFHNS invest in clinical supervision programs; therefore, understanding the efficacy of such investments is an emerging topic that warrants further research.

The implications of these findings emphasise the importance of not only the engagement of families, but the ongoing retention through to the conclusion phase. The Integrated Theory of Parental Involvement developed by Daro and McCurdy stopped short of considering the importance of closure when families prematurely terminate the working relationship with a nurse. Circumstances where families abruptly end the nurse–family partnership before goals are achieved and risk is mitigated can leave nurses with uncertainty about how best to conclude intervention. Although participants were found to be knowledgeable, confident and experienced, the practice of integrating care to safeguard children was challenging. Nurses not only require the skills and knowledge, but personal attributes are also core to safeguarding. This alone is a difficult area of practice and when considered in the context of individual factors associated with the families depicts a rich tapestry of challenges and opportunities required by the providers of nursing services to provide a meaningful response.

## 6. Conclusions

Forming and maintaining engagement of families where maltreatment is suspected require advanced nursing practices to manage the complex needs of these families. This multilevel mixed-methods study was designed using the Integrated Theory of Parent Involvement [24]; however, research findings supported the need for an extension of the framework to consider a further phase to acknowledge the conclusion of the nurse–family relationship. Operating from a universal health service model, contemporary nursing practice must extend to follow the principles of progressive universalism to ensure care is adapted to achieve this intention where a health response is warranted. This study found that establishing honesty during the initial formation of relationship between nurse and family will continue to play a critical function when the relationship is challenged by risks that warrant direct and indirect work with the family. As nurses navigate managing the needs of families, ensuring the families are retained in ongoing universal health care is an artform that relies on more than experience. Demonstrating competency across a range of skills and practices is essential in the retention phase, and confidence when working with families can result in positive outcomes for both the family and the nurse provider. The practice reality for nurses means families may abruptly disengage from their service, leaving needs unmet and nurses faced with emotional turmoil as they reconcile the unplanned conclusion of the relationship. This study found the practices of nurses are not so varied, in fact practices did not vary despite allowing for variation in engagement. As nurses continue to play a critical role for vulnerable families with children aged 0 to 5 years, the drivers that influence their practice must be adapted to ensure this continues. This is particularly so in the context of disengagement. Opportunities exist for services to consider how to enhance the use of clinical supervision, peer review and consultation to support and coach nurses to adapt their practice to maintain engagement with essential health care service, similar to the specialist service of CFHN. Nurses need support (beyond their immediate manager) to reinforce their continuous efforts to retain families in need of their service.

### 6.1. Practical Implications

This paper offers a rich description of the nuances of practice and makes recommendations that can offer value to nurses and others who work with families where violence, including child abuse, is used. There are a range of professions contributing to the preventative tier of the public health approach to child protection. The practices described are not exclusive to nursing. They may have potential application to other disciples working to keep children safe and families supported.

A core element of this practice is both the establishment and retention of meaningful connections between the nurse and family. However, these practices are complex and must be flexible to adjust to escalating needs. Nurses are required to intersperse their practice with knowledge and skills developed from experience, and professional support from within the nursing service. Each of these elements plays a valuable influence on the nurse as a provider of the service and the advanced nursing practice required to meet the complex needs of these families.

### 6.2. Contributions to the Field

The nurse–family relationship is an essential component of service delivery: a suite of strategies is needed to maintain the relationship.

Progressive universalism: this type of service places nurses in a position of profound responsibility to address the complex needs of families. Consequently, advanced nurse practice is essential to adjust the dosage of universal health services to effectively engage parents, even in circumstances where they are hard to reach. This means engagement must be flexible and nurses need to be well resourced.Applying the Model of Parent Involvement allowed for a critical examination of both the role nurses play (provider factors) but also an examination of the universal health service (programme drivers).

## Figures and Tables

**Figure 1 ijerph-20-06472-f001:**
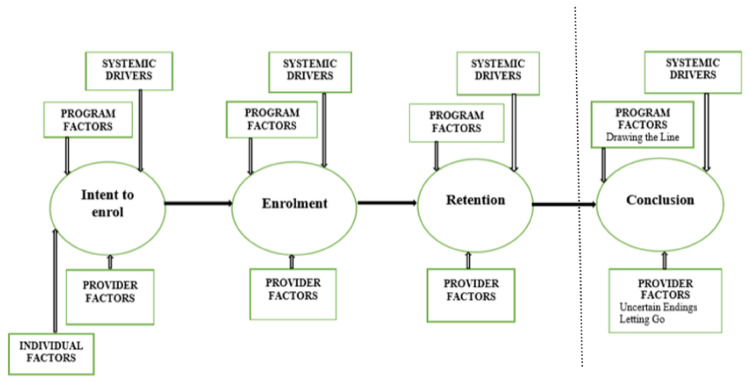
Framework of Practice for working with families with multiple and complex needs [25]. The results emphasised the role of the nurse as the provider, as this study focused on the practices of this nursing specialty. The framework extends the Integrated Theory of Parental Involvement [24] represented on the left of the divider line.

## Data Availability

The data presented in this study are available on request from the corresponding author.

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
