# Peer review of "Engagement and Retention of Families in Universal Australian Nurse-Home-Visiting Services: A Mixed-Methods Study"

_ijerph, 2023, doi:10.3390/ijerph20156472_

Round 1

Reviewer 1 Report

It was a true pleasure to read this research. Digging into the nuances is desperately needed to reveal the realities of delivering home visitation. Taking a multimethod approach with the structure of accepted models aids tackling such a complex study. 

I'd suggest tightening up on the writing as best as you can. Inefficiencies and complicated sentences make explaining a complex topic harder to understand. This is the case in the introduction (e.g., 69-76: "Understanding the nuances of nurs- 69 ing practice that enable nurses to actively engage and retain families in service delivery to 70 continue receiving nursing intervention needs urgent research attention. There has been 71 a lot written about the importance of creating and maintaining the relationship between 72 nurses and mothers in nurse home visiting models (Jack et al, 2021). But more needs to 73 be understood about the ways in which the relationship is maintained even in the face of 74 a legal and policy driven mandate to report to child protection authorities (Williams et al 75 2019)."

More importantly is attention to presenting a clear rationale and methodology for the two studies. The literature review leads to the studies' intention but I wanted to see a very clear structure and research questions for the two studies. Presenting detail on the survey - how many items and constructs, validation of the constructs, etc would help; even presenting it as an appendix. Then in the focus group, what were the key questions asked? And were they written after the analysis of the survey data or preconstructed?

Attention is given to challenges in mixed methods work, yet I was curious how the sample for the focus groups was selected. "Purposeful sampling" seems inefficient to identify who was in this sample. Given that the survey sample was heterogeneous(within a single profession in a single district), I was curious about the balance represented (e.g., all nurse educators?). 

The presentation of the results was perhaps unnecessarily complicated - perhaps by trying to be efficient. The results from both studies were presented together. Also the author's own model was presented as a framework for analysis in the results, when presented along with the McCurdy and Daro model seems to make more sense. This would give the reader a clearer sense of the model that had previously been validated. 

Perhaps its the way I'm used to reading results from two studies, but it seems clearer to present the quantitative results (albeit briefly and through tables if needed), then study two. That certain themes are repeated isn't a bad thing. This only gives more weight to the discussion which then can tie the findings together. As it is the results of the quantitative study seem to be buried, nearly leading to the question of why include it at all, and not present the meaty paper focused on focus group data. 

I wish you very well with work in this area - it is very much needed and you are tackling a lot to present such complex reality. 

Presentation of the English language was proficient. 

Reviewer 2 Report

Title: Engagement and Retention of Families in Universal Australian Nurse Home Visiting Services: A Mixed Methods Study 

Journal: Early International Journal of Environmental Research and Public Health 

Manuscript ID: IJERPH-2373248 

This article focuses on a study of 129 Child and Family Health Nurses offering universal child and family health services to families in Australia from an Integrated Theory of Parent Involvement perspective. The approach used in this multi-level, mixed methods study included the use of a survey with the full sample as well as focus groups from a subsample of participants (n = 27) to understand the relationships between program nurses and the families they serve. In phase one, a survey was used to assess nurse knowledge confidence, education, and workplace training (aligned with Provider factors of the guiding framework). Focus groups explored reporting practices, skills required to discuss concerns with families, threat of family disengagement, intention between increased contact, and concluding service delivery following parent disengagement.  

In my notes below, I outline recommendations for strengthening the manuscript. In addition, I invited a graduate student to contribute to the review with their notes added as well. Although we found the paper to have the potential to make a meaningful contribution to the field, we found the overall organization to be hard to follow, with Methods spread throughout multiple sections of the paper. With significant revision, we do believe there is meaningful content here, which could make a significant contribution field. We appreciate the authors sharing their work and hope they will continue to refine the current manuscript for publication, whether in this journal or another. Our suggestions are detailed below.  

Literature Review 

Overall, we found the literature review to be informative and thorough with wonderful insight into the aims of nurse home visiting services in Australia. A primary strength of the paper is the emphasis on home visiting as a prevention strategy and exploring the factors that lead to successful nurse-family relationships in home visiting programs over the course of program involvement and beyond.  

  1. Many different terms are introduced throughout the introduction and literature review. We recommend that authors take time to clearly define and describe each term that is introduced. For example, in paragraph 1, the authors write: “The model of care is best described as proportionate universalism (Cowley et al., 2015) or progressive universalism.” Progressive universalism was defined and used throughout this section. Is this term used interchangeably with proportionate universalism? Or should the distinction between the two be clarified? 

  1. In addition, it would be helpful to the reader to add additional details about the goals of the home visiting model described. What are the intended outcomes for families? In addition to the intended outcomes for families, what are the goals for quality/characteristics of the relationship between families and nurse providers?  

  1. Be sure to include citations for any claims made throughout the introduction/literature review section. For example, in paragraph 5, authors write: “Operating from progressive universalism enables nurses to adjust the frequency of contact with families in their care. but simply increasing contact to families is not enough. Nurses must also craft a purposeful response to child maltreatment from a universal health care service”. Please add references to support these claims.  

  1. The theoretical framework seems to Theoretical framework has several levels, including individual, provider, etc. The focus of the study, however, seems to be mostly on provider factors. We think it would be helpful for authors to mention this explicitly in the discussion as well as in the visual models provided.  

  1. Before transitioning to the methods section, it would be helpful to clearly define the study research questions. We both agreed that further additions to the literature review might be needed after adding clarity to the Methods section.  

  1. An organizational note: Paragraph 6 does not seem to flow well with the sections around it. We recommend moving paragraph 6 before the last paragraph in this section, which includes the study purpose. 

  1. And finally, we recommend moving the paragraph on ethical approval to the Methods section.  

Methods 

Our primary concerns were the paper related to the Methods section. Overall, we found the Methods section to be disorganized, with numerous overlapping sections (e.g., procedures, participants, recruitment, and measures) and without details needed to understand what was measured, validity/reliability of measures (if available), etc.  

  1. We recommend restructuring the manuscript to make sure the methods section includes some of the current sections, but also information that we found elsewhere in the manuscript. Specifically, Methods should include Participants, Procedures, Measures, analytic plan, as well as a section on how missing data were handled.  

  1. In the procedures section, please include details on how, when, where participants filled out questionnaires and the specific procedures for focus groups, including questions asked and approach taken in asking questions/probing.  

  1. In the measures section, please include all information about each construct, how they were measured (sample questions), what participants, and whether or not there are any measures of validity/reliability associated with these measures.  

  1. In the article, the authors note that (Page 4): “The primary author drew on a social work lens...”  Please say more about what this means and how this might have impacted study design, methodological choices, as well as interpretation of results. 

Results 

The results section had a wealth of information included, but some of the information would have been important to include in the Methods section instead.  

  1. A note on - Table 1: the lines and green color make the table complicated for the reader to interpret. When we first examined the table, it appeared that the first two rows belong to the enrolment stage. While scanning the discussion, we realized only the first row belongs to the enrolment stage. 

  1. There are inconsistencies in the use of percentages, sometimes given in parentheses (93%, n=120), sometimes not.   

  1. Authors presented the results in three phases: enrolment, retention, and conclusion. It was unclear how these three phases aligned with the two phases of research design (they were not integrated). Moreover, it was hard to understand as a reader what the confidence and frequency scales are (the confidence scale is in the second paragraph of 4.1, and the frequency scale is in the third paragraph of 4.2).  

  1. Quantitative and qualitative results were not presented in a way that was organized or clear for the reader to understand. Moreover, results seemed to be blended across the Results and Discussion/Conclusion sections. Organizing around research questions as well as across survey / focus group following a more streamlined and thorough methods section would help clarify the aims of the paper as well as the findings in a meaningful way. 

  2. Thank you for the opportunity to review this manuscript. We hope that author(s) find our comments to be helpful in strengthening the submission, if not for publication in this journal, then to pursue publication in others. Thank you for your work!   

N/A.

Reviewer 3 Report

The manuscript titled “Engagement and Retention of Families in Universal Australian  Nurse Home Visiting Services: A Mixed Methods Study” examined the capacity to engage and retain families, including those at risk for child maltreatment and family violence. 

-The abstract does not provide an informative and balanced summary of what was done and what was found. Phrases like "Child and Family Health Nurses (n=129) 12 participated in a pragmatic, multilevel mixed-methods study using the McCurdy and Daro (2001) Integrated Theory of Parent Involvement" are unnecessary in the abstract. It should focus on providing a coherent summary of the most significant findings and conclusions.

-It is important for authors to ensure that the introduction is more concise, focused, and follows a clear structure. The introduction of the manuscript appears to be overly redundant and challenging for readers to follow seamlessly. To address this concern, authors should consider adopting a more structured approach that guides the reader through the relevant scientific background, emphasizes the significance of the work, and clearly presents the hypothesis. The overall clarity and specificity of the research objectives and hypotheses are lacking.By incorporating an additional paragraph in the concluding section of the introduction, the authors can rectify this issue. 

The paragraph 2, "Background to the study design," could benefit from simplification and potentially be incorporated into the Method section

-A major concern lies in the lack of clarity and detail regarding the methodology employed in the study. It is essential for authors to provide a comprehensive and transparent explanation of the procedures, protocols, and data collection techniques utilized. By doing so, readers can better understand the study's design and methodology, facilitating replication of the findings. The authors have not adequately described the methodology in a manner that enables a clear understanding of the scientific method employed. The absence of standardized measures, exclusion/inclusion criteria, and statistical analyses undermines the scientific validity of the results.

-Another significant issue observed in the manuscript is the authors' failure to adhere to the journal's bibliographic guidelines

Round 2

Reviewer 2 Report

Thank you to the authors for taking the time to thoughtfully incorporate feedback from the reviewers. The flow and clarity of the manuscript has been significantly improved since the initial submission.